# Repression of CHROMOMETHYLASE 3 prevents epigenetic collateral damage in *Arabidopsis*

**Ranjith K Papareddy[1]†\*, Katalin Páldi[1]†, Anna D Smolka[1], Patrick Hüther[1,2], Claude Becker[1,2], Michael D Nodine[1,3]\***

[1]Gregor Mendel Institute (GMI), Austrian Academy of Sciences, Vienna Biocenter (VBC), Dr. Bohr-Gasse 3, Vienna, Austria; [2]Genetics, LMU Biocenter, Ludwig-Maximilians University, Martinsried, Germany; [3]Laboratory of Molecular Biology, Wageningen University, Wageningen, Netherlands

**Abstract** DNA methylation has evolved to silence mutagenic transposable elements (TEs) while typically avoiding the targeting of endogenous genes. Mechanisms that prevent DNA methyltransferases from ectopically methylating genes are expected to be of prime importance during periods of dynamic cell cycle activities including plant embryogenesis. However, virtually nothing is known regarding how DNA methyltransferase activities are precisely regulated during embryogenesis to prevent the induction of potentially deleterious and mitotically stable genic epimutations. Here, we report that microRNA-mediated repression of CHROMOMETHYLASE 3 (CMT3) and the chromatin features that CMT3 prefers help prevent ectopic methylation of thousands of genes during embryogenesis that can persist for weeks afterwards. Our results are also consistent with CMT3-induced ectopic methylation of promoters or bodies of genes undergoing transcriptional activation reducing their expression. Therefore, the repression of CMT3 prevents epigenetic collateral damage on endogenous genes. We also provide a model that may help reconcile conflicting viewpoints regarding the functions of gene-body methylation that occurs in nearly all flowering plants.

**\*For correspondence:**
rpapareddy1@g.ucla.edu (RKP);
michael.nodine@wur.nl (MDN)

†These authors contributed equally to this work

**Competing interests:** The authors declare that no competing interests exist.

## Introduction

Methylation of DNA encoding transposable elements (TE) is required to silence their expression and consequently prevent them from mobilizing and mutagenizing genomes (*Kato et al., 2003*; *Law and Jacobsen, 2010*). Complex mechanisms have evolved to balance the high degree of sensitivity needed to direct methylation and silencing of TEs with the precision required to prevent ectopic methylation of endogenous genes (*Antunez-Sanchez et al., 2020*; *Ito et al., 2015*; *Papareddy et al., 2020*; *Saze and Kakutani, 2011*; *Williams et al., 2015*; *Zhang et al., 2020*). However, little is known about the mechanisms of epigenome homeostasis during embryogenesis when organisms are particularly vulnerable to TE-induced mutagenesis, as well as the establishment of potentially deleterious epimutations that can persist through many cell divisions and even across generations (*Henderson and Jacobsen, 2007*; *Mathieu et al., 2007*; *Probst et al., 2009*; *Saze et al., 2003*; *Mittelsten Scheid et al., 1998*).

In *Arabidopsis thaliana* (Arabidopsis), most TEs are found in pericentromeric regions of the genome to which RNA polymerases have limited access (*Arabidopsis Genome Initiative, 2000*; *Lippman et al., 2004*; *Zhang et al., 2006*). These TE-enriched constitutive heterochromatic regions are characterized by high densities of cytosine methylation in symmetric (CG or CHG; H ≠ G) and asymmetric (CHH) contexts, as well as histone H3 lysine dimethylation (H3K9me2) and other transcriptionally repressive chromatin marks (*Cokus et al., 2008*; *Lister et al., 2008*; *Stroud et al.,*

*2014*). Symmetric DNA methylation and H3K9me2 also facilitate the stable propagation of silenced states through cell divisions (*Jackson et al., 2002*; *Lindroth et al., 2001*; *Stroud et al., 2013*, *Stroud et al., 2014*). METHYLTRANSFERASE 1 (MET1) maintains CG methylation through mitotic and meiotic cell divisions with high fidelity due to VARIANT IN METHYLATION 1/2/3 (VIM1/2/3) proteins that recognize hemi-methylated CG and recruit MET1 to methylate daughter strands (*Feng et al., 2010*; *Finnegan and Dennis, 1993*; *Ning et al., 2020*; *Woo et al., 2008*). CG methylation can also recruit RNA Polymerase IV complexes required to produce 24-nt small interfering RNAs (siRNAs) that are then loaded onto Argonaute proteins and guide them to target loci by base-pairing with nucleic acids (*Blevins et al., 2015*; *Herr et al., 2005*; *Papareddy et al., 2020*; *Zhai et al., 2015*; *Zilberman, 2003*). This leads to the recruitment of DOMAINS REARRANGED METHYLTRANSFERASES 1/2 (DRM1/2) and results in de novo methylation of cytosines in all sequence contexts, including CHH, which is a hallmark feature of RNA-directed DNA methylation (RdDM) (*Cao and Jacobsen, 2002*; *Stroud et al., 2013*; *Wierzbicki et al., 2008*). However, RdDM is typically restricted from constitutive heterochromatin because it is inaccessible to DNA-dependent RNA polymerase IV and methyltransferases required for RdDM (*Papareddy et al., 2020*; *Zemach et al., 2013*). Instead, CHH methylation of constitutive heterochromatin is mediated by CHROMOMETHYLASE 2 (CMT2) that binds to H3K9me2 deposited by KRYPTONITE (KYP) and closely related SUPPRESSOR OF VARIEGATION 3–9 HOMOLOGUE PROTEIN 5/6 (SUVH5/6) methyltransferases (*Stroud et al., 2014*; *Zemach et al., 2013*). CHROMOMETHYLASE 3 (CMT3) also forms interlocking positive feedback loops with H3K9 methyltransferases (*Du et al., 2012*; *Jackson et al., 2002*; *Lindroth et al., 2001*) but is more closely associated with the cell cycle and mediates CHG methylation (*Ning et al., 2020*).

CMT3-mediated CHG methylation is largely deposited on TEs. However, CMT3 can also induce the ectopic methylation of protein-coding genes (*Wendte et al., 2019*). Moreover, the introduction of Arabidopsis CMT3 transgenes into *Eutrema salsugineum*, which lost CMT3 millions of years ago (*Bewick et al., 2016*), could occasionally reconstitute CG methylation on genes (*Wendte et al., 2019*). The resulting gene-body methylation (gbM) could be stably maintained independent of the CMT3 transgene for several generations (*Wendte et al., 2019*). However, it remains largely unknown how CMT3 is restricted to targeting heterochromatin, as well as the consequences of CMT3-induced hypermethylation of genes. Moreover, the functional significance of gbM in animals and plants has been intensely debated. Because methylated cytosines are mutagenic due to associated cytosine deamination (*Shen et al., 1992*; *Sved and Bird, 1990*), features associated with gbM have been interpreted as evidence that gbM provides selective advantages that counterbalance this mutagenesis-imposed fitness penalty. For instance, gbM is depleted from transcription start and end sites (*Tran et al., 2005*; *Zhang et al., 2006*; *Zilberman et al., 2007*), and it has recently been reported that gbM helps prevent transcription initiation from cryptic promoters located in gene bodies as initially proposed (*Choi et al., 2020*; *Zilberman et al., 2007*). Moreover, gbM tends to be enriched on constitutively expressed genes (*Lister et al., 2008*; *Niederhuth et al., 2016*; *Takuno et al., 2017*; *Zhang et al., 2006*), which would be consistent with gbM stabilizing gene expression by excluding certain histone variants (i.e. H2A.Z) from genes (*Coleman-Derr and Zilberman, 2012*) and generally enhancing gene expression (*Muyle and Gaut, 2019*; *Shahzad et al., 2021*). Nevertheless, accumulating evidence is also consistent with gbM being a heritable by-product of CMT3-induced epimutations (*Bewick et al., 2016*; *Bewick et al., 2019*; *Wendte et al., 2019*).

Consistent with the need to fine-tune the amount of CMT3 activities required to both silence TEs and prevent epimutations on genes, mechanisms exist that transcriptionally (*Ning et al., 2020*) and post-translationally (*Deng et al., 2016*) regulate CMT3, as well as remove H3K9me2 specifically from expressed genes (*Inagaki et al., 2010*; *Saze et al., 2008*). These and additional mechanisms are likely of utmost importance during embryo development when a proliferative morphogenesis phase produces the most fundamental cell lineages of the plant, including those that will eventually generate the gametes. Yet, how DNA methylation pathways are regulated during this phase of dynamic cell division to exquisitely balance the need for TE methylation with the prevention of potentially deleterious and stably inherited epimutations is virtually unknown.

## Results

### Cell division is linked with CG and CHG methylation through distinct mechanisms

MET1 and VIM1/2/3 are required for the faithful transmission of mCG across cell cycles (*Feng et al., 2010*; *Finnegan and Dennis, 1993*; *Ning et al., 2020*; *Woo et al., 2008*) and accordingly had increased transcript levels in rapidly dividing early embryos that also correlated well with transcripts encoding cell-cycle activators throughout embryogenesis (*Figure 1A*; *Hofmann et al., 2019*; *Papareddy et al., 2020*). More specifically, MET1 and VIM1/2/3 transcript levels peaked at the early heart stage and were reduced afterwards before plummeting at the mature green stage. These transcript developmental dynamics were also characteristic of transcripts encoding proteins involved in licensing DNA replication (e.g. Cyclins A2/B1, CDKB1-1, MINICHROMOSOME MAINTENANCE2), heterochromatin maintenance (e.g. DDM1) and DNA methylation (e.g. CMT3), but not randomized controls (*Figure 1B*, *Figure 1—figure supplement 1A*). Therefore, genes required for DNA methylation and heterochromatin maintenance are tightly correlated with cell-cycle activity during embryogenesis.

To test whether the patterns observed for transcripts regulating DNA methylation reflect DNA methylation dynamics, we computed differentially methylated cytosines (DMCs) across flowers, embryos and leaves (see Materials and methods). Similar to previous observations (*Bouyer et al., 2017*; *Lin et al., 2017*; *Papareddy et al., 2020*), 70% of DMCs occurred in the CHH context (*Figure 1C*). Consistent with dynamic expression patterns of MET1 and CMT3, substantial fractions of DMCs occurred in CG (20%) or CHG (10%) contexts, respectively. Therefore, DNA methylation is dynamically reconfigured in all sequence contexts during embryogenesis. In total, these symmetric DMCs represented 1185 CG (*Supplementary file 1*) and 1398 CHG (*Supplementary file 2*) differentially methylated regions (DMRs) covering 201 kb and 185.8 kb, respectively (*Figure 1D*, *Supplementary file 1*; see Materials and methods). Although a significant fraction of CG and CHG DMRs overlapped (n = 183; 7.1% of total), the vast majority of CG and CHG DMRs were located in non-overlapping genomic regions corresponding to euchromatic gene-rich and heterochromatic TE-rich regions of the genome, respectively (*Figure 1D*). Because CHG methylation can require CG methylation (*Stroud et al., 2013*), we tested whether the 15.1% of CHG DMRs overlapping CG DMRs require CG methylation. Leaves deficient in CG methylation did not have reduced CHG methylation in CHG DMRs regardless of whether or not they overlapped with CG DMRs (*Figure 1E,F*; data from *Stroud et al., 2013*). This indicates that CHG DMRs occur in distinct genomic regions and are largely independent of CG methylation.

Relative to floral bud samples, CG DMRs have slightly reduced methylation in preglobular embryos, followed by increased methylation until after the torpedo stage, when levels dramatically reduce in mature green embryos and recover in leaves (*Figure 1G*). By contrast, methylation levels of CHG DMRs are relatively stable between floral buds and early embryos, then decrease in late embryos, reaching a minimum in leaves (*Figure 1H*). Accordingly, changes in CG and CHG DMR methylation levels during development were significantly correlated with MET1 (Pearson's R = 0.8; p value = 0.03) and CMT3 (Pearson's R = 0.74; p value = 0.05) transcript levels, respectively (*Figure 1—figure supplement 1B,C*). Therefore, although cell division rates are correlated with symmetric DNA methylation dynamics, distinct mechanisms reconfigure CG or CHG methylation genome-wide during embryogenesis.

### Genome-wide coordination of symmetric DNA methylation

Because DNA methylation is concentrated on TEs (*Stroud et al., 2013*; *Zhang et al., 2006*), we next investigated global developmental dynamics of TE methylation. CG methylation on both euchromatic and heterochromatic TEs was slightly reduced in pregobular embryos and then restored to the levels found in floral buds by the early heart stage (*Figure 2A,B*). Whereas CG methylation of euchromatic TEs was relatively constant for the remainder of embryogenesis, heterochromatic TEs had significantly increased methylation during late embryogenesis compared to post-embryonic tissues. Consistent with heterochromatin becoming highly condensed during embryo maturation (*van Zanten et al., 2011*), we found that CG hypermethylation in mature green compared to bent cotyledon embryos predominantly occurred in pericentromeric genomic regions rather than gene-

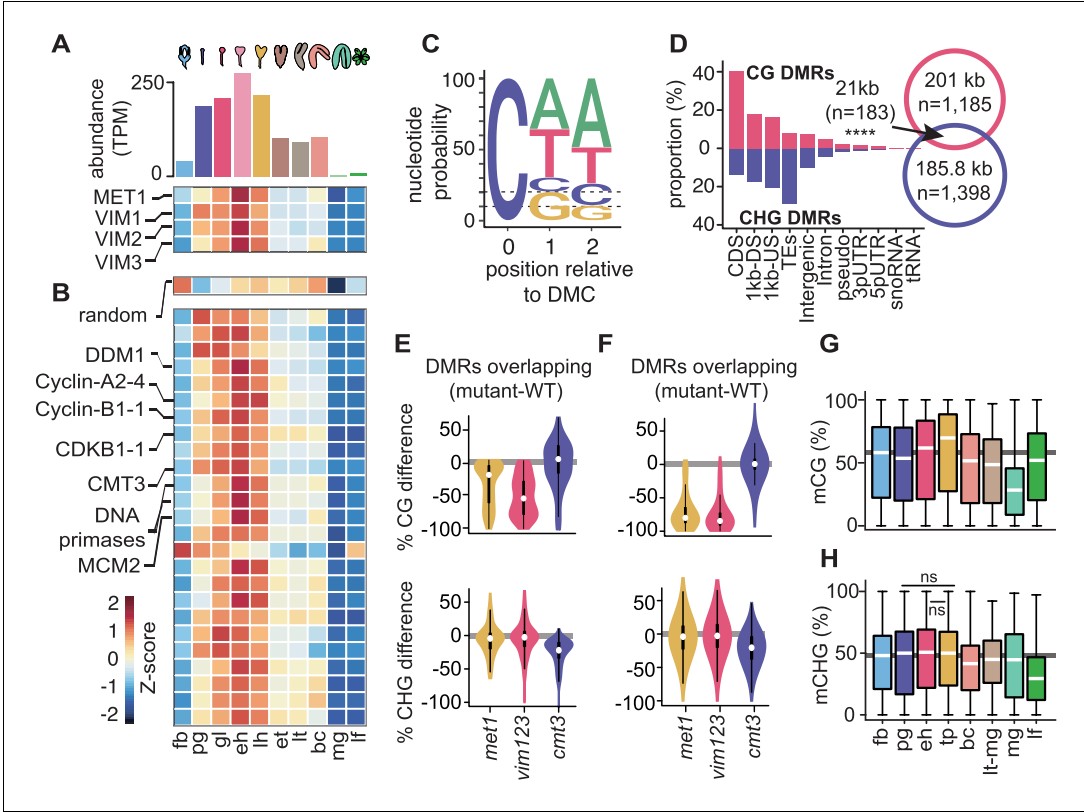

**Figure 1.** Cell division is linked with CG and CHG methylation through distinct mechanisms. (**A**) Bar chart depicting total abundance (*top*) and heat map of individual relative transcript levels (*bottom*) of genes involved in CG methylation in three biological replicates each of flowers, embryos, and leaves (*Hofmann et al., 2019*). fb, floral buds; pg, preglobular; gl, globular; eh, early heart; lh, late heart; et, early torpedo; lt, late torpedo; bc, bent cotyledon; mg, mature green; lf, leaf. (**B**) Heatmap showing developmental dynamics of permuted gene set (*top*) median values (i.e. 1000 iterations of random sampling of 25 genes) and top-25 genes co-varying with MET1, VIM1, VIM2, and VIM3 obtained by employing nearest neighbor algorithm calculated based on Euclidean distance between genes and centroid expression of MET1, VIM1, VIM2, and VIM3 (*bottom*). (**C**) Sequence logo representing nucleotide probability relative to differentially methylated cytosines (DMC). (**D**) Proportion of CG and CHG differentially methylated regions (DMRs) overlapping genomic features. Venn diagram showing overlap between CG and CHG DMRs. Significance overlap of DMRs determined by Fisher's Exact test p value < 0.0001 is indicated by ****. (**E and F**) Violin plot showing CG (*top*) and CHG (*bottom*) methylation differences between mutant and WT leaves for CHG DMRs overlapping (**E**) or not overlapping (**F**) with CG DMRs (*Stroud et al., 2013*). (**G and H**) Box plots of average weighted methylation of CG DMRs (n = 1185) and (**G**) CHG DMRs (n = 1398) during development. fb, floral buds; pg, preglobular; eh, early heart; tp, torpedo (6 DAP) *Pignatta et al., 2015*; bc, bent cotyledon; lt-mg, late torpedo-to-early mature green *Hsieh et al., 2009*; mg, mature green *Bouyer et al., 2017*; lf, leaf. fb, pg, eh, bc and lf were from *Bouyer et al., 2017*; *Hsieh et al., 2009*; *Papareddy et al., 2020*; *Pignatta et al., 2015*. Unless stated as not significant (ns), all combinations are significant with p values < 0.001 obtained by Mann-Whitney U test. Shaded horizontal line in the background represents the median methylation value of floral buds.

The online version of this article includes the following figure supplement(s) for figure 1:

**Figure supplement 1.** Characteristics of genes and differentially methylated regions co-expressed with symmetric methyltransferases.

rich chromosomal arms (*Figure 2C*). CG methylation was required for the production of 24-nt siRNAs from euchromatic TEs, but only marginally for heterochromatic TEs (*Figure 2—figure supplement 1A*, data from *Lister et al., 2008*). Conversely, the loss of 24-nt siRNAs in *nrpd1a* mutants only had negligible effects on CG methylation of both heterochromatic and euchromatic TEs (*Figure 2—figure supplement 1B,C*). Therefore, siRNA production from euchromatic regions of the genome requires CG methylation, but not vice versa.

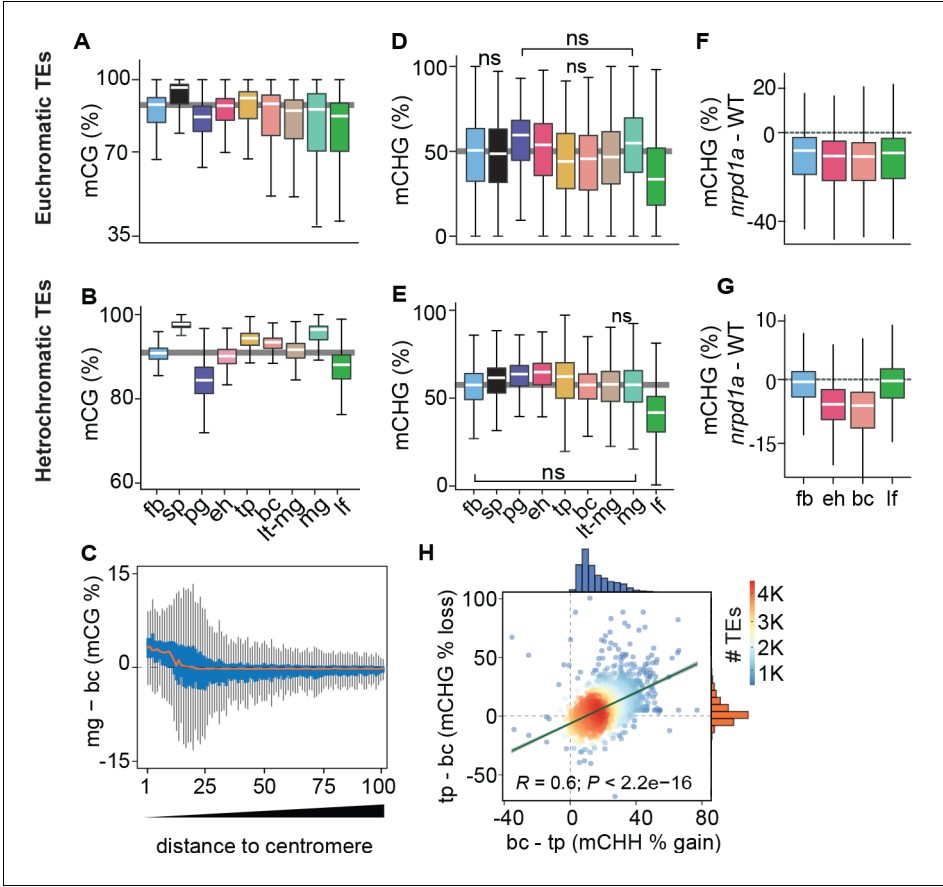

**Figure 2.** Genome-wide coordination of symmetric DNA methylation. (**A and B**) Boxplots of CG methylation percentages on euchromatic (**A**) and heterochromatic (**B**) TEs during development. fb, floral buds; sp, sperm (*Ibarra et al., 2012*); pg, preglobular; eh, early heart; tp, torpedo (6 DAP); bc, bent cotyledon; lt-mg, late torpedo-to-early mature green; mg, mature green; lf, leaf. Thick horizontal bars indicate medians, and the top and bottom edges of boxes represent the 75th and 25th percentiles, respectively. Shaded horizontal line in the background represents the median methylation value of floral buds. (**C**) Difference in CG methylation between mature green (mg) and bent cotyledon (bc) embryos were calculated in 1 kb genomic bins, which were divided into percentiles and sorted based on their distance to centromeres (1 and 100 being the tile closest and furthest from centromeres, respectively). Red color line indicates the median and the top and bottom edges of the blue colored boxes represent 75th and 25th percentiles, respectively. Vertical gray bars indicate 1.5X the interquartile range. (**D and E**) Boxplots of CHG methylation on euchromatic (**D**) and heterochromatic (**E**) TEs during development (key as in **A**). (**F and G**) Boxplots of CHG methylation differences between *nrpd1a* and WT (Col-0) tissues for euchromatic (**F**) and heterochromatic (**G**) TEs. (**H**) Scatterplot showing Pearson's correlation coefficients (**R**). Differences in mCHH and mCHG between bent cotyledon (bc) and torpedo stage (tp) embryos are shown on x- and y-axes, respectively. Histograms show the number of TEs in thousands (**K**).

The online version of this article includes the following figure supplement(s) for figure 2:

**Figure supplement 1.** Relationships between MET1 and 24-nt siRNAs.

Global CHG methylation of euchromatic and heterochromatic TEs was higher in embryos compared to leaves (*Figure 2D,E*). Similar to previous observations for CHH methylation (*Papareddy et al., 2020*), siRNA-deficient *nrpd1a* mutant tissues had reduced CHG methylation on euchromatic or heterochromatic TEs in all or only embryonic samples, respectively (*Figure 2F,G*). Intriguingly, increased CHH methylation on heterochromatic TEs was significantly correlated with decreased CHG methylation during late stages of embryogenesis when cell division rates are reduced (*Figure 2H*). Therefore, CMT3-dependent CHG and CMT2-dependent CHH methylation of heterochromatic TEs are positively and negatively correlated with cell division rates, respectively.

## Repression of CMT3 during embryogenesis regulates methylome dynamics

CMT3 is recruited to loci by binding to H3K9me2 deposited by SUVH4/5/6 histone methyltransferases (*Du et al., 2012*; *Jackson et al., 2002*; *Lindroth et al., 2001*; *Stroud et al., 2014*). CMT3 and KYP, which is the major SUVH4 H3K9 methyltransferase, were dynamically expressed according to patterns characteristic of other cell-cycle regulated genes and CHG methylation dynamics (Figure 1B,H, *Figure 3A*). More specifically, CMT3 and KYP were highly expressed in rapidly dividing early embryos and had reduced expression in late embryos until the mature stage, where they were barely detectable. Altogether, our results are consistent with the idea that the more rapid cell divisions in early embryos demand higher levels of CMT3 and KYP to maintain mCHG through the cell cycle. Moreover, IBM1, which encodes an H3K9me2 demethylase and prevents CMT3 recruitment to gene bodies (*Miura et al., 2009*; *Saze et al., 2008*), is dynamically expressed during embryogenesis in a pattern that strongly resembles CMT3 and KYP (*Figure 3A*). Therefore, co-expression of IBM1 with CMT3 and KYP likely helps limit ectopic H3K9me2 and methylated CHG on gene bodies during

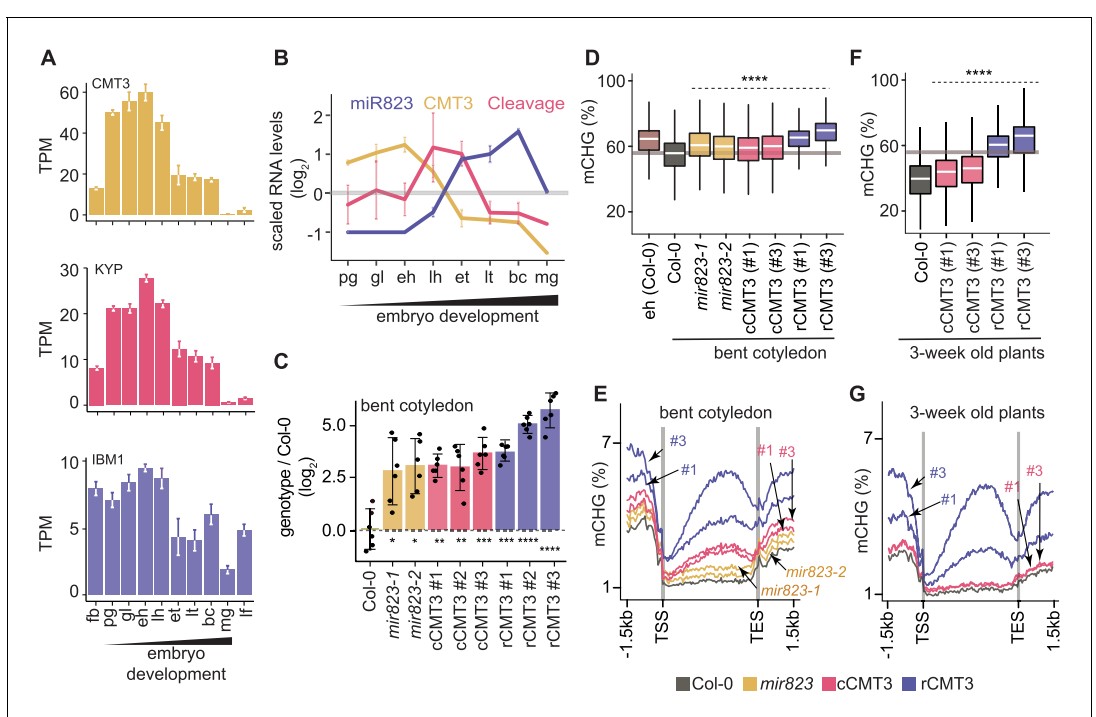

**Figure 3.** Repression of CMT3 during embryogenesis regulates methylome dynamics. (A) Barplots illustrating transcript levels of CMT3 (*top*), KYP (*middle*), and IBM1 (*bottom*) in flowers, embryos, and leaves. fb, floral buds; pg, preglobular; gl, globular; eh, early heart; lh, late heart; et, early torpedo; lt, late torpedo; bc, bent cotyledon; mg, mature green; lf, leaf. (B) Line graphs showing the relative RNA abundance of miR823 (*blue*), CMT3 RNA (*yellow*), and miR823:CMT3 cleavage products (*pink*). (C) Log₂-transformed relative CMT3 target transcript levels in bent cotyledon embryos (8 DAP; day after pollination) from WT plants (Col-0), or *cmt3-11* plants expressing either miR823-cleavable CMT3 (cCMT3) or miR823-resistant CMT3 (rCMT3) versions. Each dot represents the mean of two technical replicates of embryos and bars represent mean values. Error bars in A-C represent standard errors of the means of three biological replicates. Asterisks indicate whether the transcript levels observed in *mir823* mutant, cCMT3 and rCMT3 embryos were significantly different compared to WT (Two-tailed Student's t tests; ****, ***, **, and * represent p values < 0.0001, < 0.001, < 0.01, and < 0.05, respectively). Color-coded according to the key. (D) Boxplots of CHG methylation on transposons with ≥five informative cytosines covered by ≥four reads and classified as either euchromatic or heterochromatic in *Papareddy et al., 2020*. pValues < 0.0001 based on Mann-Whitney U tests of methylation differences between WT and either mutant or transgenic bent cotyledon embryos are represented by ****. (E) Metaplots of average CHG methylation percentages across genes bodies from transcription start sites (TSS) to transcription end sites (TES), 1.5 kb upstream and 1.5 kb downstream of genes in bent cotyledon embryos. Color-coded according to the key. (F and G) Boxplots of CHG methylation on transposons (F) and metaplots of CHG methylation on genes (G) in three-week old plants as described in D and E, respectively.

The online version of this article includes the following figure supplement(s) for figure 3:

**Figure supplement 1.** *mir823* mutants and effects of miR823-directed repression of CMT3.

embryogenesis as has been demonstrated during post-embryonic development (*Inagaki et al., 2017*).

We previously found that miR823-directed cleavage of CMT3 transcripts is highly enriched in embryos directly after morphogenesis (*Plotnikova et al., 2019*). In contrast to CMT3 transcript dynamics, miR823 accumulates during embryogenesis, and miR823:CMT3 cleavage products were enriched and significantly detected specifically at late heart and early torpedo stages precisely when CMT3 transcript levels were sharply decreasing (*Figure 3B*). Based on these observations, we hypothesized that miR823-mediated repression of CMT3 contributes to the reduced CHG methylation levels observed during late embryogenesis.

To test if miR823-directed repression of CMT3 transcripts reduces CHG methylation levels during embryogenesis, we generated deletions in the region of the *MIR823* locus encoding the mature miRNA (*Figure 3—figure supplement 1A*) and examined CMT3 transcript and CHG methylation levels. Both independently generated *mir823-1* and *mir823-2* mutants were confirmed as nulls (*Figure 3—figure supplement 1B*) and had significantly increased CMT3 levels relative to wild type (WT) in embryos at the bent cotyledon stage when CMT3 levels are normally reduced (*Figure 3C*). Consistent with miR823-directed cleavage of CMT3 being highly enriched in embryos, we did not observe increased CMT3 transcripts in either leaves or floral buds of *mir823* mutants (*Figure 3—figure supplement 1D*). Moreover, CHG, but not CG or CHH, methylation was increased on TEs in bent cotyledon embryos of both *mir823-1* and *mir823-2* mutants relative to WT (*Figure 3D*, *Figure 3—figure supplement 1E*).

As an independent approach, we used site-directed mutagenesis to introduce synonymous mutations in the miR823 target site within CMT3 transgene constructs that included 1.41 kb upstream and 0.73 kb downstream intergenic regions, and associated cis-regulatory elements (*Figure 3—figure supplement 1C*; see Materials and methods). As controls, we also generated CMT3 constructs without mutations, and introduced these miR823-cleavable CMT3 (cCMT3), as well as the miR823-resistant (rCMT3), constructs into *cmt3-11* mutant plants (*Henderson and Jacobsen, 2008*). CMT3 transcript levels were increased in rCMT3 relative to cCMT3 lines at the bent cotyledon stage (*Figure 3C*), but not in leaves or floral buds (*Figure 3—figure supplement 1D*), which further indicates that miR823-directed cleavage and repression of CMT3 is highly enriched in embryos transitioning between morphogenesis and maturation. CMT3 levels were also increased in cCMT3 and rCMT3 lines compared to Col-0 in embryos, leaves, and floral buds (*Figure 3C*, *Figure 3—figure supplement 1D*) suggesting that miR823 is not sufficient to repress transgenic CMT3 to the same extent as endogenous CMT3 transcripts. Although we cannot rule out that this is due to missing cis-regulatory repressive elements in the transgenes, increased gene dosage and positional effects of the transgenes seems more likely. Upstream and downstream intergenic regions were included in the CMT3 constructs (*Figure 3—figure supplement 1G*). Moreover, although relative transgene copy numbers were not significantly different across the independently generated cCMT3 and rCMT3 transgenic lines, they were higher than endogenous CMT3 in WT (*Figure 4—figure supplement 1G,H*). Nevertheless, it is clear that CMT3 levels are finely tuned during embryogenesis. Together with the analysis of *mir823* mutants and miR823-mediated CMT3 transcript cleavage products (*Plotnikova et al., 2019*), these results strongly indicate that miR823 cleaves and represses CMT3 levels during mid-embryogenesis. Consistent with what we observed in *mir823* mutants, increased CMT3 transcript levels in cCMT3 and rCMT3 embryos resulted in CHG hypermethylation of TEs (*Figure 3D*) but did not globally influence CG or CHH methylation (*Figure 4—figure supplement 1E,F*). Remarkably, increased CMT3 transcript levels in *mir823* mutants, cCMT3 and most strikingly rCMT3 embryos were associated with ectopic CHG methylation on protein-coding gene bodies and flanking regions in bent cotyledon embryos (*Figure 3E*). Therefore, both TEs and genes are hypermethylated when CMT3 levels are not properly downregulated upon the morphogenesis-to-maturation transition during embryogenesis.

To test whether miR823-directed repression of CMT3 and prevention of CHG methylation of genes that we observed in embryos persists after embryogenesis, we next profiled methylomes of cCMT3 and rCMT3 plants 3 weeks after germination. We chose to focus on rCMT3 plants because of the large amount of hypermethylation observed in these lines during embryogenesis, and used cCMT3 plants as controls. Although TEs had increased CHG methylation levels in both cCMT3 lines relative to WT, protein-coding genes were not affected (*Figure 3F,G*). In stark contrast, TEs and genes were hypermethylated in both rCMT3 lines compared to cCMT3 or WT plants, and only

slightly reduced relative to the levels observed in rCMT3 bent cotyledon embryos (*Figure 3F,G*). Together with miR823-independent processes (e.g. IBM1 removal of H3K9me2), miR823-directed repression of CMT3 is therefore required to prevent the hypermethylation of protein-coding genes that can be maintained weeks after the completion of embryogenesis.

## Chromatin features associated with CMT3-induced gene methylation

To yield insights into how genes are hypermethylated upon the derepression of CMT3, we determined whether certain genomic features were associated with CMT3-induced genic methylation. Toward this end, we first selected 22,637 nuclear-encoded protein-coding genes that had ≥5 methylC-seq reads overlapping CHG sites in rCMT3 line #3 and that were expressed (i.e. ≥1 TPM in any tissue based on *Hofmann et al., 2019*). We chose rCMT3 line #3 because it had the strongest genome-wide CHG hypermethylation and focussed on expressed genes to exclude those that may have TE-like features, which could confound analysis. We then used k-means clustering of the differences between rCMT3 line #3 and WT bent cotyledon embryos to partition this set of genes into four clusters (*Figure 4—figure supplement 1A*). These clusters were comprised of 1439–7882 genes (6.4–34.8% of total) and ranged from groups of genes that had no methylation changes (cluster 1) to those that were strongly hypermethylated with 3′ biases (cluster 4) in rCMT3 compared to WT embryos (*Figure 4A,B*, *Figure 4—figure supplement 1B,C*). The same patterns were observed across these clusters in embryos from an independently generated rCMT3 transgenic (line #1), which indicates that CMT3-induced hypermethylation is not stochastic (*Figure 4—figure supplement 1B, C*).

TE-like methylated (teM) genes generally have non-CG methylation on their gene bodies without strong 3′ biases (*Kawakatsu et al., 2016*; *Bewick et al., 2016*). To check whether rCMT3-induced genic CHG methylation is affected by teMs, we intersected our gene clusters with previously defined teMs (*Bewick et al., 2016*) and found that only 272 of 22,637 (0.012%) expressed genes overlapped teMs (*Figure 4—figure supplement 1D*). Hypermethylated clusters (clusters 3 and 4) contained more teMs compared to unmethylated or lowly methylated gene clusters (*Figure 4—figure supplement 1E*). However, rCMT3 embryos still had genic CHG hypermethylation and 3′ biases after excluding teM genes, whereas, WT embryos remained devoid of CHG methylation (*Figure 4—figure supplement 1F,G*). Therefore, our analysis is not confounded by either TEs or teM genes. As expected, CMT3-induced hypermethylation predominantly occurred in the CMT3-preferred CWG context (*Gouil and Baulcombe, 2016*; *Li et al., 2018*) although hypermethylation was also found in CCG and slightly, but significantly in non-CHG contexts, including CG characteristic of gbM similar to previous observations (*Figure 4—figure supplement 1H*; *Wendte et al., 2019*).

Consistent with methyltransferases preferring nucleosome-rich DNA as substrates (*Chodavarapu et al., 2010*; *Du et al., 2012*), CMT3-induced hypermethylation was proportional to patterns of nucleosome occupancy and biased towards the 3′ ends of gene bodies, which was highly similar to CG methylation (*Figure 4C*, *Figure 4—figure supplement 1I*). Nucleosome spacing is promoted by linker histone 1 (H1) (*Choi et al., 2020*; *Fan et al., 2003*) and CMT3-induced CHG hypermethylation was proportional to H1 levels across gene bodies (*Figure 4—figure supplement 1J*). Because nucleosome occupancy was not as readily distinguishable between clusters of affected genes (i.e. clusters 2–4) (*Figure 4—figure supplement 1I,J*), we hypothesized that histone variants conferring differential nucleosome stabilities and chromatin accessibility may influence ectopic CMT3-induced hypermethylation (*Osakabe et al., 2018*). Indeed, CHG hypermethylation across the four groups was positively correlated with levels of the stable histone variants H2A, H2A.X and most notably H2A.W that was recently shown to be required for CHG methylation (*Figure 4E,F*, *Figure 4—figure supplement 1K*; *Bourguet et al., 2021*; *Yelagandula et al., 2014*). CMT3-induced CHG hypermethylation was also tightly associated with transcriptionally repressive H3K9me2 marks, which are required for interdependent feedback loops with CMT3 (*Figure 4G*). It was inversely related to H2A.Z (*Figure 4H*) and marks indicative of active transcription including H3K4me3 and H3K9ac (*Figure 4I*, *Figure 4—figure supplement 1L*). Further suggesting that deregulated CMT3 prefers features typically associated with inaccessible chromatin, genes with CHG hypermethylation had reduced chromatin accessibility (*Figure 4—figure supplement 1M*) and were generally closer to heterochromatic centromeres (*Figure 4—figure supplement 1N*). Moreover, the most CMT3-induced hypermethylated genes (i.e. cluster 4) were also substantially hypermethylated in *ddm1* mutants (*Figure 4J*) that have increased heterochromatic accessibility (*Figure 4K*) and decreased

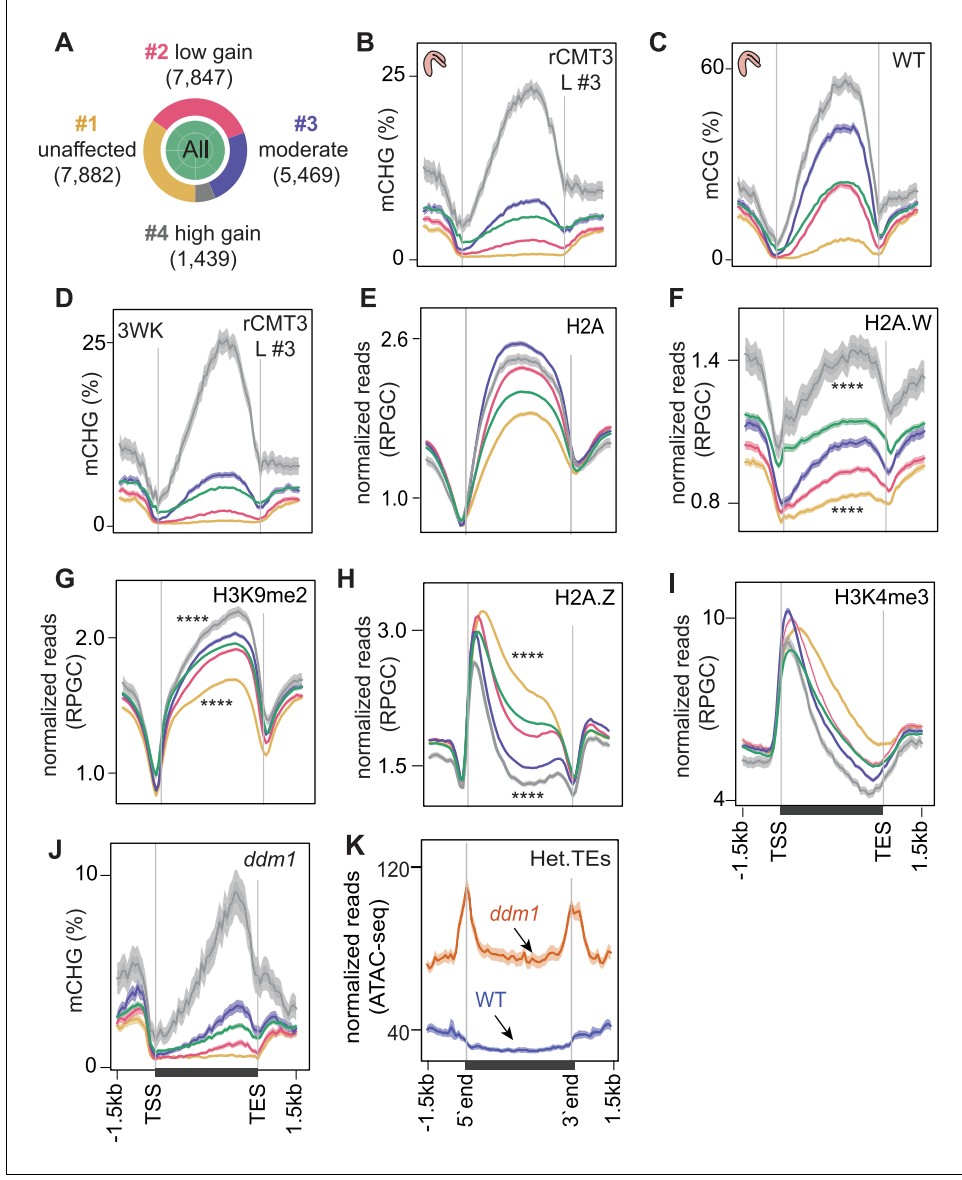

**Figure 4.** Chromatin features associated with CMT3-induced gene methylation. (**A**) Proportion of genes in each cluster partitioned using k-means clustering algorithm based on differences in mCHG between rCMT3 (line #3) and WT embryos. Unaffected genes (*yellow*), low mCHG gain genes (*red*), moderate mCHG gain genes (*blue*), and high mCHG gain genes (*gray*). Green inner circle represents all expressed genes. (**B–D**) Metaplots showing mCHG on gene clusters in bent cotyledon embryos from rCMT3 line #3 (L #3) (**B**), mCG on gene clusters in WT bent cotyledon embryos (**C**) and mCHG in rCMT3 (L #3) 3-week-old plants (3WK) (**D**). Shaded ribbons in metaplots represent standard deviations. (**E–I**) Metaplots showing normalized reads per genomic content (RPGC) average values of histone variant H2A (**E**), H2A.W (**F**) (*Yelagandula et al., 2014*), H3K9me2 (**G**) (*Stroud et al., 2014*), H2A.Z (**H**) (*Yelagandula et al., 2014*), and H3K4me2 (**I**) (*Maher, 2020*). p Values < 0.0001 obtained by Mann-Whitney U test based on differences between genes in cluster 1 or four compared to all genes is represented by ****. (**J**) Metaplots showing mCHG on gene clusters in seventh generation *ddm1* mutants (*Stroud et al., 2013*). (**K**) Normalized ATAC-seq reads (*Zhong et al., 2021*) representing accessibility of heterochromatic TEs (Het.TEs) in WT and *ddm1* mutants as defined in *Papareddy et al., 2020*.

The online version of this article includes the following figure supplement(s) for figure 4:

**Figure supplement 1.** Partitioning of CMT3-induced hypermethylated genes and associated chromatin features.

stability (*Mathieu et al., 2003*; *Soppe et al., 2002*; *Zhong et al., 2021*). Although CMT3-induced CHG hypermethylation was strongly associated with CG gene-body methylation (gbM), both the independence of developmental mCHG DMRs (*Figure 1E,F*) and the gain of mCHG being associated with proportional loss of mCG over genes in *ddm1* mutants (*Figure 4—figure supplement 1O*; *Ito et al., 2015*; *Stroud et al., 2013*; *Zemach et al., 2013*) indicate that mCG is not strictly required for ectopic CHG hypermethylation of genes. Instead, the associations between chromatin features of genes and their propensity for CMT3-induced hypermethylation altogether suggest that excessive CMT3 is ectopically recruited to genic chromatin characterized by nucleosome stability and inaccessibility.

## Impact of CMT3-induced hypermethylation on gene expression

Because CHG methylation of TEs contributes to their repression (*Stroud et al., 2014*), we tested whether CMT3-induced ectopic CHG hypermethylation of protein-coding genes also represses their expression levels. Namely, we performed mRNA-seq on three biological replicates of WT and rCMT3 (line #s 1 and 3) bent cotyledon embryos. Principal component analysis revealed that WT and rCMT3 biological replicates clustered according to genotype and in similar positions along the dominant principal component axis corresponding to developmental time (*Figure 5A*). This indicates that our mRNA-seq datasets captured gene expression variation inherent to WT and rCMT3 genotypes, as well as that our staging was accurate. Differences in global transcript levels were not observed across the four clusters with increasing levels of CMT3-induced CHG methylation suggesting that ectopic CHG methylation alone was not sufficient to globally repress gene expression (*Figure 5B*, *Figure 5—figure supplement 1A*). We then identified 916 genes that were differentially expressed between rCMT3 and WT embryos (i.e. $\geq$2-fold differences and adj. p values $\leq$ 0.01; see Materials and methods) (*Figure 5—figure supplement 1B,C* and *Supplementary file 3*). Differentially expressed genes (DEGs), defined by comparing either rCMT3 line #1 or rCMT3 line #3 with WT, were commonly detected in both independently generated lines with 87.5% of genes overlapping (*Figure 5—figure supplement 1C*). In both rCMT3 lines, DEGs were less hypermethylated compared to all expressed genes, which indicates that the vast majority of changes in gene expression observed upon up-regulation of CMT3 were not directly due to their hypermethylation (*Figure 5—figure supplement 1D*). We then examined whether hypermethylation affects a subset of genes by computing DMRs in rCMT3 compared to WT bent cotyledon embryos and identified 4603 (97% of total) and 127 (3% of total) CHG hypermethylated and hypomethylated DMRs, respectively (*Supplementary file 4*; see Materials and methods). Further suggesting that CHG hypermethylation has minimal direct consequences on the expression of most genes under the conditions examined, we found that only a small but significant number of the down-regulated genes (including 1.5 kb regions flanking their transcriptional units) overlapped DMRs (21 of 542, 3.8% of total; Fisher's exact test, p value = 1.29e-05) (*Figure 5—figure supplement 1E*). Consistent with CMT3-induced hypermethylation repressing their expression, the DMRs overlapping these 21 down-regulated genes were significantly CHG hypermethylated compared to genomic bins (*Figure 5C*). Moreover, the stronger-expressing rCMT3 line #3 had significantly higher CHG methylation compared to rCMT3 line #1 (*Figure 5C*). This further supports that increased CMT3 levels lead to more ectopic CHG methylation (*Inagaki et al., 2010*; *Inagaki et al., 2017*). However, the transcript levels of these 21 genes were only moderately reduced in rCMT3 line #3 compared to rCMT3 line #1, suggesting non-linear relationships between gene hypermethylation and transcript levels (*Figure 5—figure supplement 1F*). Strikingly, transcripts corresponding to these 21 CMT3-induced hypermethylated and down-regulated genes were rapidly increasing when embryos were transitioning to the maturation phase (*Figure 5D*). Moreover, 10 of these 21 genes (Fisher's exact test, *P* value = 1.49e-13), were among a group of 381 genes previously identified to also be rapidly activated at these time points (*Figure 5—figure supplement 1G*; *Hofmann et al., 2019*). Nearly half of these 381 genes (n = 183, 48%) were also among the 563 significantly down-regulated genes in rCMT3 embryos compared to WT. Altogether, our expression and methylation analyses suggest that when CMT3 is not properly repressed it can induce ectopic hypermethylation of genes. Furthermore, we suggest that CMT3-induced hypermethylation of gene promoters or bodies can reduce the steady state levels of transcripts from genes that are in the process of switching from silent to active transcriptional states. However, additional experiments are required to directly test whether gene-body CHG methylation can repress gene expression.

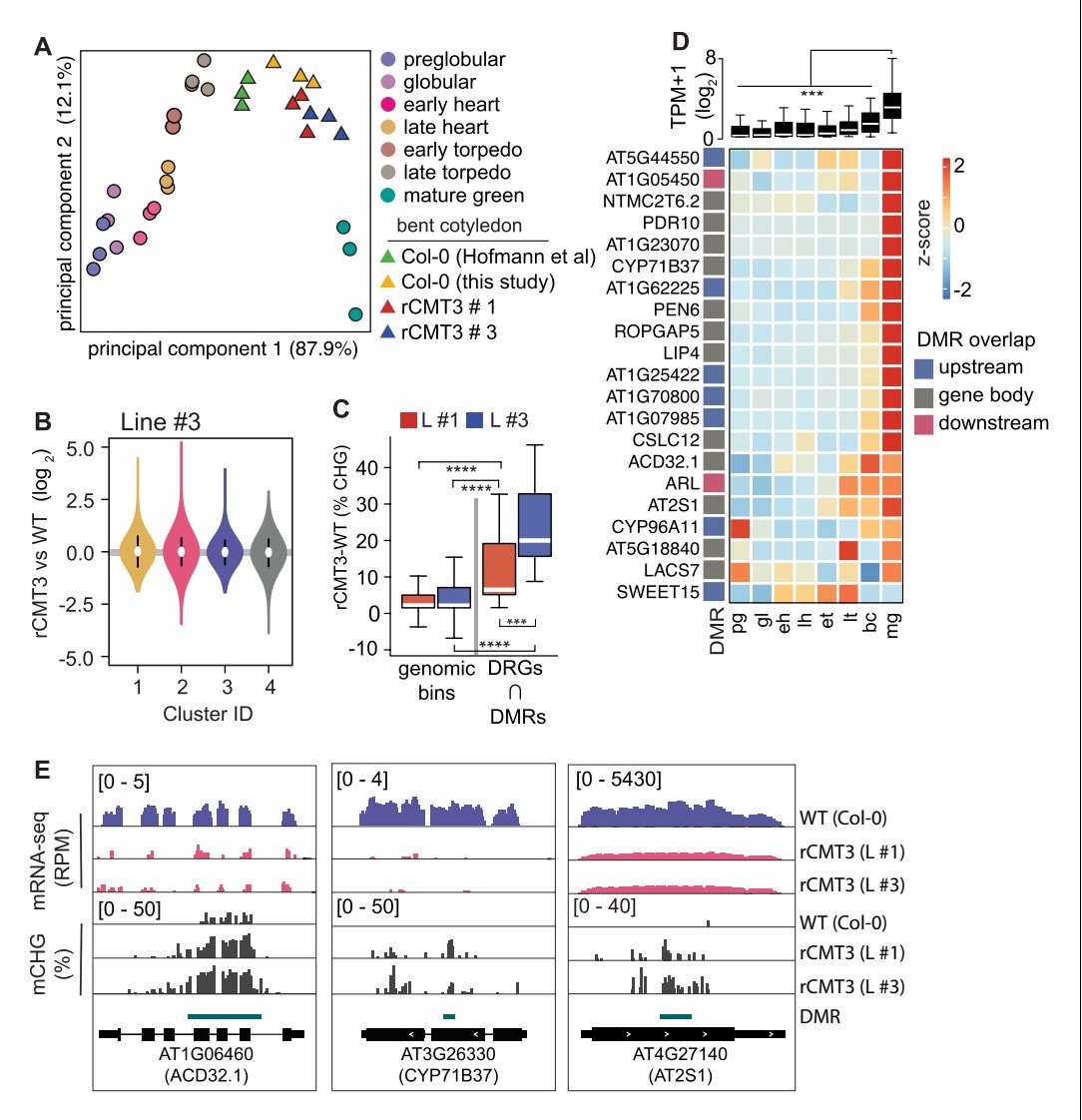

**Figure 5.** Impact of CMT3-induced hypermethylation on gene expression. (**A**) Principal component analysis of mRNA-seq from three biological replicates of rCMT3 and WT (Col-0) bent cotyledon embryos generated in this study along with floral buds, embryos, leaves, from *Hofmann et al., 2019* and color-coded according to the key. (**B**) Violin plot showing transcript fold changes in rCMT3 (line #3) compared to WT (Col-0) bent cotyledon embryos per cluster as defined in *Figure 4A*. (**C**) Boxplot showing difference in methylation comparing rCMT3 to WT in down-regulated genes (DRG) intersecting with DMRs and similarly sized genomic bins of 213 bp as controls. p Values *< 0.001 and* < 0.0001 based on Mann-Whitney U tests are represented by *** and ****, respectively. (**D**) Boxplot (*top*) and heatmap (*bottom*) of transcript levels of DRGs intersecting DMRs during embryogenesis. p Values < 0.001 based on differences in transcript levels between mature green (mg) and all other stages of embryogenesis based on Mann-Whitney U test are represented by ***. (**E**) Integrative genome viewer (IGV) screenshot of representative downregulated genes associated with DMRs.

The online version of this article includes the following figure supplement(s) for figure 5:

**Figure supplement 1.** Additional information regarding the influence of CMT3-induced hypermethylation on gene expression.

## Discussion

DNA methylation is faithfully propagated across cell cycles by methyltransferases to ensure robust silencing of TEs (*Borges et al., 2021*; *Law and Jacobsen, 2010*; *Mathieu et al., 2007*; *Ning et al., 2020*; *Probst et al., 2009*; *Saze et al., 2003*). However, it is not well understood how DNA

methyltransferases are regulated following periods of rapid division to prevent off-targeting of genes and their consequential repression. Cell division rates are highly dynamic during *Arabidopsis* embryogenesis. We found that the expression of MET1 and CMT3 methyltransferases and corresponding CG and CHG methylation are intricately linked to mitotic indices through distinct mechanisms (*Figure 1*). Moreover, miR823-mediated cleavage and repression of CMT3 following the proliferative early phase of embryogenesis helps prevent excess CMT3 from ectopically methylating protein-coding genes that can persist for weeks afterwards (*Figure 3*). CMT3-induced hypermethylation of genes was highly associated with features conferring nucleosome stability (*Figure 4*) and resulted in the repression of genes that are transcriptionally activated (*Figure 5*). Repression of CMT3 following a period when it is needed in high quantity to keep pace with TE methylation therefore prevents CMT3 from ectopically targeting protein-coding genes for methylation. This resulting epigenetic collateral damage on protein-coding genes appears to negatively affect gene expression. Our results are consistent with the model that CMT3-induced epimutations give rise to CG gene-body methylation (gbM) that can be maintained by MET1 across many generations (*Wendte et al., 2019*).

Complex mechanisms are required to specifically silence mutagenic TEs rather than endogenous genes (*Antunez-Sanchez et al., 2020*; *Deng et al., 2016*; *Lee et al., 2021*; *Lister et al., 2008*; *Papareddy et al., 2020*; *Saze and Kakutani, 2011*; *Williams et al., 2015*; *Zhang et al., 2020*). Mechanisms regulating epigenome homeostasis are of paramount importance during *Arabidopsis* embryogenesis due to highly dynamic cell cycle and transcriptional activities, as well as the establishment of cell lineages that will produce all future cell types including the gametes. MET1 and CMT3 methyltransferases are required for TE methylation (*Kato et al., 2003*; *Stroud et al., 2014*) and are expressed at high levels during early embryogenesis likely because this is a period of rapid cell division. CHG and CHH methylation exhibit opposite developmental dynamics depending on the tissue's mitotic index (*Figure 1*, *Figure 2*, *Figure 6*; *Papareddy et al., 2020*). When embryos are transitioning to stages with reduced cell division, decreased CMT3-mediated CHG methylation is correlated with increased CMT2-mediated CHH methylation (*Figure 2H*). Unlike CMT2, CMT3 can also target protein-coding genes for CHG methylation (*Stroud et al., 2014*) and lead to the recruitment of transcriptionally repressive H3K9me2 methyltransferases such as KYP (*Du et al., 2014*; *Jackson et al., 2002*; *Lindroth et al., 2001*). Therefore, handing over TE silencing to CMT2-dependent CHH methylation in cells with reduced division rates likely reduces ectopic methylation of protein-coding genes. In addition to what we observed during embryogenesis, varying degrees of mitotic indices across development can readily explain the genome-wide patterns of non-CG methylation reported thus far (*Borges et al., 2021*; *Calarco et al., 2012*; *Gutzat et al., 2020*; *Ji et al., 2019*; *Kawakatsu et al., 2016*, *Kawakatsu et al., 2017*; *Lin et al., 2017*; *Narsai et al., 2017*; *Papareddy and Nodine, 2021*).

CMT3, KYP and their corresponding DNA and histone methylation marks form interdependent feedback loops that perpetuate silencing through cell divisions (*Du et al., 2015*; *Ning et al., 2020*). Consistent with the transcription-coupled H3K9me2 demethylase IBM1 breaking these loops and preventing ectopic CHG hypermethylation of genes, we found that CMT3, KYP, and IBM1 were highly expressed during early embryogenesis (*Figure 3*). After this rapidly dividing morphogenesis phase, transcripts from CMT3, KYP, and IBM1 decrease, and miR823 directs the cleavage and repression of excess CMT3 to help prevent hypermethylation of protein-coding genes (*Figure 3*). Excess CMT3 induces CHG methylation on distinct regions of protein-coding genes that are characteristic of stable nucleosomes including transcriptionally repressive H3K9me2 marks that bind to CMT3. Although the distribution of CMT3-induced CHG hypermethylation is strikingly similar to CG gene-body methylation of genes (*Figure 4B,C*), this appears to be due to common targeting mechanisms by CMT3 and MET1 rather than a strict prerequisite of CG. In fact, mutants with reduced CG methylation (*Figure 4—figure supplement 1O*; *Jacobsen and Meyerowitz, 1997*; *Lister et al., 2008*; *Saze and Kakutani, 2007*; *Stroud et al., 2013*) or species largely devoid of genic CG methylation (*Wendte et al., 2019*) can still recruit CHG on genes. CMT3-induced CHG methylation of genes that we observed in rCMT3 transgenic plants was similar to ectopic gain of genic mCHG in *ddm1* mutants (*Figure 4*). Notably, heterochromatin becomes destabilized in *ddm1* mutants (*Figure 4K*; *Mathieu et al., 2003*; *Soppe et al., 2002*) and CMT3 prefers features associated with stable (*Figure 4E,F*, *Figure 4—figure supplement 1K*; *Bourguet et al., 2021*; *Osakabe et al., 2018*; *Yelagandula et al., 2014*) over unstable nucleosomes such as H2A.Z with active marks

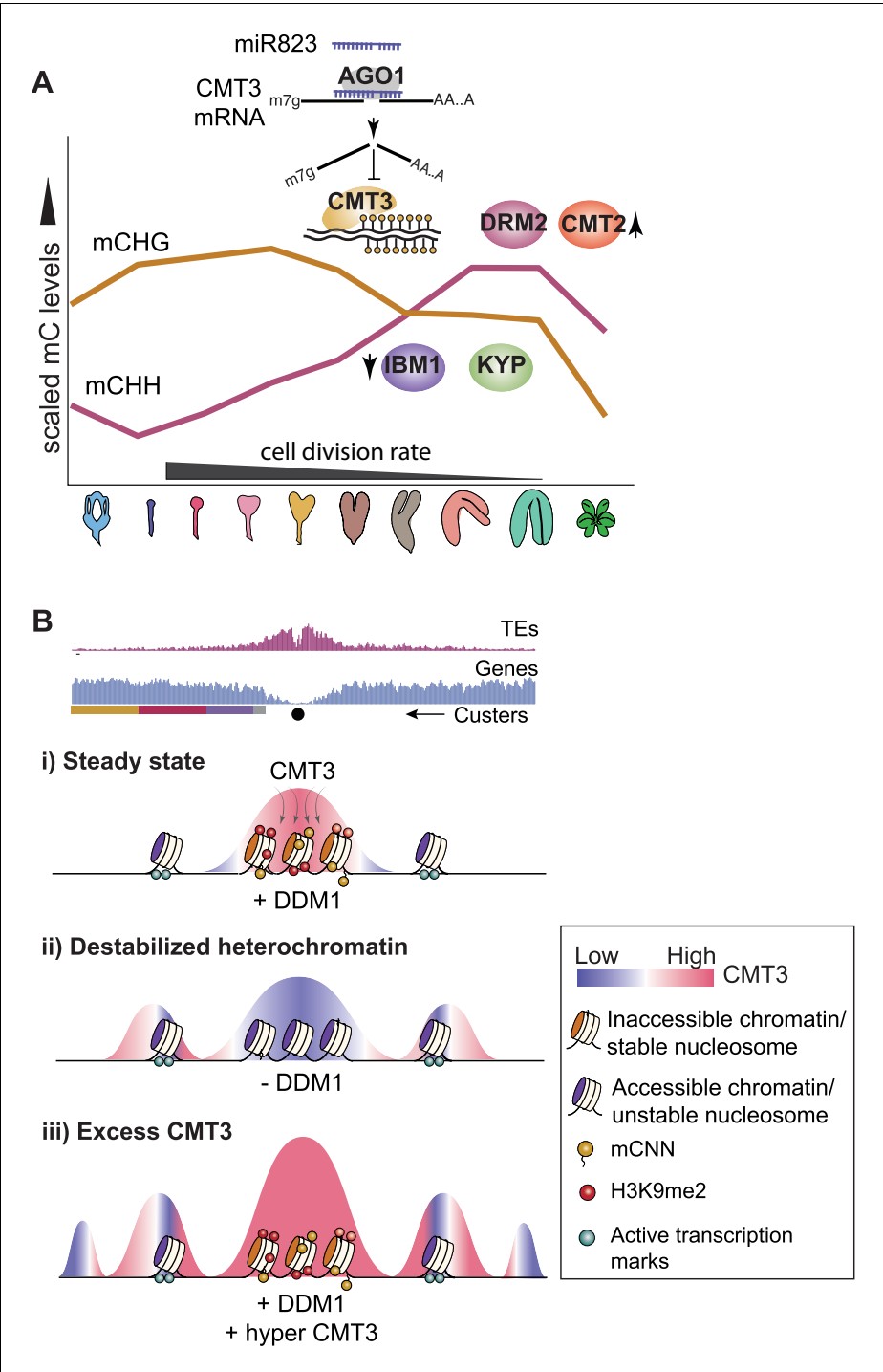

**Figure 6.** Models for CMT3 regulation during periods of fluctuating cell division rates and destabilized chromatin. (**A**) Model of non-CG methylation dynamics during embryo development and corresponding regulatory mechanisms. (**B**) Model for how CMT3 equilibrium is maintained to restrict its activity to heterochromatin. Density of transposable elements (TEs) (*top; red*) and genes (*middle; blue*) on chromosome 1. Cartoon illustration of gene cluster location (*bottom*) according to key in **Figure 4A**. Black dot represents the centromere. (i) In steady state, stable nucleosomes along with H3K9me2 and DNA methylation provides positive reinforcement to sequester CMT3 to constitutive heterochromatin. (ii) Loss of DDM1 results in destabilized and accessible heterochromatin (**Zhong et al., 2021**), characterized by loss of H3K9me2 and stable nucleosomes (**Osakabe et al., 2021**). Accessible chromatin or DNA without stable nucleosomes is no longer a preferable substrate for CMT3 and results in CHG hypomethylation of TEs. CMT3 will now be readily available and redirected to genic regions where it

*Figure 6 continued on next page*

*Figure 6 continued*

induces ectopic CHG methylation in proportion to the levels of stable nucleosomes and chromatin marks. (iii) Excess levels of CMT3 causes genome-wide CHG hypermethylation with a preference for stable nucleosomes associated with repressive marks that tend to be in regions closer to centromeres compared to chromosomal arms.

(*Figure 4H,I*, *Figure 4—figure supplement 1L*). Although destabilization of heterochromatin has been inversely correlated with genic CHG methylation (*Ito et al., 2015*; *Zhang et al., 2020*), chromatin features underlying this mechanism are unclear. Therefore, we propose that destabilization of heterochromatin in *ddm1* mutants redirects CMT3 to genic regions with stable nucleosomes. Therefore, factors such as DDM1 that stabilize heterochromatin may be yet another mechanism required to regulate CMT3 activity in order to achieve proper epigenome homeostasis (*Figure 6*).

CMT3-induced CHG hypermethylation of genes did not globally affect steady state transcript levels (*Figure 5*). However, we observed exceptional association between CHG hypermethylation and repression of genes that switch from transcriptionally inactive to active states. Because IBM1-mediated removal of H3K9me2 marks is coupled to transcription (*Inagaki et al., 2017*), it is possible that CMT3-induced methylation can form feedback loops with H3K9me2 methyltransferases when genes are transcriptionally inert. However, when genes are switched on, H3K9me2 could repress initial rounds of transcription before it is removed by IBM1. Accordingly, it may be difficult to detect the effects of ectopic CHG methylation on gene expression when quantifying transcripts at steady state with standard mRNA-seq. It is possible that we observed a repressive effect of CHG hypermethylation on a subset of genes because we profiled a developmental stage in which hundreds of genes become transcriptionally activated at the onset of embryo maturation. Nevertheless, we cannot completely exclude that the repression of hypermethylated genes undergoing transcriptional activation is due to secondary effects of other genes influenced by CMT3-induced hypermethylation. Importantly, CMT3-induced CHG hypermethylation due at least partially to loss of miR823 repression in embryos is largely maintained for weeks after detectable miRNA activity (*Figure 3*). Therefore, epigenetic collateral damage occurring in embryos may also negatively impact gene expression later in life. However, additional experiments are required to directly test the relationship between CMT3-induced hypermethylation and gene expression.

Transcriptional (*Ning et al., 2020*), post-transcriptional (*Figure 3*), post-translational (*Deng et al., 2016*), post-hoc (*Saze et al., 2008*) and perhaps substrate-related (*Figure 4*) mechanisms fine-tune CMT3 activities to levels required to specifically silence mutagenic TEs but not genes. However, errors in restricting CMT3 to heterochromatin are inevitable on an evolutionary timescale (*Zhang et al., 2020*) and recent studies indicate that CMT3-induced methylation of genes precedes gbM (*Wendte et al., 2019*). Because gbM can be stably maintained over many generations by MET1 and its functional significance is debatable (*Bewick et al., 2016*; *Bewick et al., 2019*; *Choi et al., 2020*; *Coleman-Derr and Zilberman, 2012*; *Le et al., 2020*; *Picard and Gehring, 2017*; *Shahzad et al., 2021*; *Takuno and Gaut, 2013*; *Wendte et al., 2019*; *Williams et al., 2021*; *Zilberman, 2017*), it cannot be excluded that gbM is merely an evolutionary record of epigenetic collateral damage events that occurred in the past (*Bewick and Schmitz, 2017*; *Bewick et al., 2017*). Our results suggest that derepressed CMT3 and MET1 both prefer genic regions characterized by increased nucleosome stability (*Figure 4*). Accordingly, CMT3-induced CHG hypermethylation tends to occur away from transcription start and end sites of genes in a nearly identical pattern as observed for gbM (*Figure 4*). We propose that CHG methylation is more tolerated in central/3' biased regions because they are relatively inaccessible to trans-acting factors that regulate transcription. Moreover, our results tentatively suggest that CMT3-induced hypermethylation can repress genes that are transcriptionally activated (*Figure 5*). Perhaps genes that are consistently expressed can accumulate CHG methylation without having a large effect on steady state transcript levels and resulting fitness penalties, and thus be more likely to accumulate gbM over evolutionary time. In other words, miR823-mediated repression is one of several ways to prevent CMT3 from ectopically methylating protein-coding genes. However, CMT3 off-targeting on genes may still occur despite these complex regulatory mechanisms and the resulting epigenetic collateral damage can be recorded as heritable gbM. The characteristic features of gbM may not pertain to its current

functions, but rather the consequences of transient CHG methylation that occurred in the past and were selected on during evolution.

## Materials and methods

**Key resources table**

| Reagent type (species) or resource | Designation | Source or reference | Identifiers | Additional information |
|---|---|---|---|---|
| Gene (*Arabidopsis thaliana*) | CHROMOMETHYLASE 3 (CMT3) | TAIR | AT1G69770 | |
| Gene (*Arabidopsis thaliana*) | MICRORNA 823A (MIR823A) | TAIR | AT3G13724 | |
| Genetic reagent (*Arabidopsis thaliana*) | miR823-cleavable CMT3 (cCMT3) | this paper | | pAlligatorR43/promoterCMT3::genomicCMT3 |
| Genetic reagent (*Arabidopsis thaliana*) | miR823-resistant CMT3 (rCMT3) | this paper | | pAlligatorR43/promoterCMT3::resistantCMT3 (generated from cCMT3 with site-directed mutagenesis) |
| Recombinant DNA reagent | pAlligatorR43 (plasmid) | DOI:10.7554/eLife.04501 | | mCherry selection marker |
| Recombinant DNA reagent | pHSE401 (plasmid) | Addgene | #62201 | CRISPR/Cas9 plasmid |
| Recombinant DNA reagent | pCBCD-T1T2 (plasmid) | Addgene | #50590 | CRISPR/Cas9 plasmid |
| Strain (*Arabidopsis thaliana*) | *cmt3-11T* | NASC | SALK_148381 | T-DNA insertion mutant of *CMT3* |
| Strain (*Arabidopsis thaliana*) | *mir823-1* | this paper | | miR823 knockout mutant |
| Strain (*Arabidopsis thaliana*) | *mir823-2* | this paper | | miR823 knockout mutant |
| Commercial kit | Q5 Site-Directed Mutagenesis Kit | New England Biolabs | #E0554S | |
| Commercial kit | Fast SYBR Green Master Mix | Roche | #06402712001 | |
| Commercial kit | SuperScript III Reverse Transcriptase | Thermo Fisher Scientific | #18080093 | |
| Commercial kit | TRIzol | Invitrogen | #15596026 | |
| Software | Lightcycler 96 | Roche Diagnostics | Version 1.1.0.1320 | |

### Plant material and growth conditions

*Arabidopsis thaliana* accession Columbia-0 (Col-0) were grown in controlled growth chambers at 20–22°C under a 16 hr light/8 hr dark cycle with incandescent lights (130 to 150 μmol/m$^2$/s).

### Generation of transgenic lines

The control genomic CMT3 construct (miR823-cleavable; cCMT3) was generated by PCR amplification of the CMT3 locus including 1408 bp upstream and 730 bp downstream of the TAIR10-annotated transcription start and end sites, respectively. PCR primers included overhangs for subsequent Gibson assembly into MultiSite-Gateway destination vector pAlligatorR43 (*Kawashima et al., 2013*). The miR823-resistant CMT3 construct (rCMT3) was generated by PCR site-directed mutagenesis (Q5 Site-Directed Mutagenesis Kit, New England Biolabs) using the cCMT3 construct as a template to introduce six silent mutations as shown in *Figure 3—figure supplement 1C*. Both cCMT3 and rCMT3 construct sequences were analyzed for mutations using Sanger sequencing. All primers used are listed in the *Supplementary file 5*. The constructs were transformed into *cmt3-11T* (SALK_148381) using the Agrobacterium floral dip method (*Clough and Bent, 1998*), and

transformants were selected based on seed-coat RFP signal under fluorescent light (Zeiss SteREO DiscoveryV.8). Multiple independent first-generation transgenic (T1) lines were identified for cCMT3 and rCMT3, and three and four were characterized in bent cotyledon embryos for each, respectively.

## Generation of CRISPR/Cas9 knockout mutants for *MIR823*

CRISPR/Cas9 knockout mutants in *MIR823* were created by using a modified pHSE401 binary vector (Addgene #62201) according to the protocol detailed by *Xing et al., 2014*. Primers containing the sequences for the two guide RNAs targeting the *MIR823* locus flanking the miR823 sequence (*Figure 3—figure supplement 1A* and *Supplementary file 5*) were amplified together with the pCBCD-T1T2 plasmid (Addgene #50590), and the resulting PCR product was subsequently assembled into the pHSE401 binary vector using GoldenGate cloning method (*Xing et al., 2014*). Plants were transformed with the floral dip method as described above; and Cas9-positive seeds were selected based on the presence of seed coat RFP signal. Deletion lines were identified with PCR using primers flanking gRNA-targeted sites (*Figure 3—figure supplement 1A* and *Supplementary file 5*). Deletion mutants were confirmed and mapped by Sanger sequencing.

## qRT-PCR analysis

Leaves (two-week old rosettes), floral clusters (five weeks) and bent cotyledon embryos (eight DAP) were homogenized in 500 µl TRIzol reagent (Invitrogen) and total RNA was isolated and purified according to manufacturer's recommendations. For mRNA, 200 ng of total RNA was used for cDNA synthesis with SuperScript III Reverse Transcriptase (Thermo Fisher Scientific). The cDNA was diluted two-fold for embryos or ten-fold for leaves and floral buds with nuclease-free water. Two µL of diluted cDNA was used as a template for the qRT-PCR with Fast SYBR Green Master Mix (Roche) on a LightCycler 96 instrument (Roche) with two technical replicates for each biorep. For miRNA823 quantification, corresponding stem-loop primers were added to the RT reaction (adapted from *Yang et al., 2014*) and miR823 levels were measured using Fast SYBR Green Master Mix (Roche) with miRNA823-specific forward primer and a stem-loop specific universal reverse primer. U6 snRNA was used as the reference RNA (adapted from *Shen et al., 2010*). Primers used for qRT-PCR are listed in *Supplementary file 5*.

## Sample size estimation, embryo isolation, and nucleic acid extraction

Sample sizes were determined based on a combination of the required statistical power, ability to acquire samples and cost of the experiments. Bent cotyledon embryos were dissected from seeds 8 days after pollination and also selected based on morphology to ensure accurate staging. Embryos were serially washed 4× with nuclease-free water under an inverted microscope. Approximately 50 embryos per replicate were isolated and stored at −80°C until further use. RNA was isolated as previously described (*Lutzmayer et al., 2017*; *Plotnikova et al., 2019*). Genomic DNA was extracted from embryos and 3-week old plants using *Quick*-DNA Micro prep Kit (Zymo D3020) according to the recommendations of the manufacturer.

## DNA methylation profiling and analysis

MethylC-Seq libraries were generated as described previously (*Papareddy et al., 2020*) and sequenced in single-read mode on an Illumina HiSeq 2500 or Nextseq 550 instrument. Adapters and the first six bases corresponding to random hexamers used during the pre-amplification step were trimmed from MethylC-seq reads using *Trim Galore*. Bisulfite-converted reads were aligned against the TAIR10 genome (*Lamesch et al., 2012*) in non-directional mode using *Bismark* (bismark −non_−directional -q −score-min L,0,–0.4) (*Krueger and Andrews, 2011*). *Methylpy* software was used to extract weighted methylation rates for each available cytosine from BAM files containing only deduplicated and uniquely mapped reads (*Schultz et al., 2015*). Reads mapping to the unmethylated chloroplast genome were used to calculate bisulfite conversion rates. FASTQ files obtained from publicly available methylomes generated from sperm (*Ibarra et al., 2012*), early torpedo (*Pignatta et al., 2015*), mid-torpedo to early maturation (*Hsieh et al., 2009*), mature green embryos (*Bouyer et al., 2017*) and DNA methylation mutant leaves (*Stroud et al., 2013*) were also processed in a similar manner except that alignments were performed in directional mode and only 5′ end

nucleotides of the reads with m-bias were removed. Differentially methylated regions (DMRs) were identified using *Methylpy* (*Schultz et al., 2015*). Briefly, two biological replicates were pooled and differentially methylated cytosines (DMCs) were identified by root mean squared tests with false discovery rates ≤ 0.01. DMRs were defined by collapsing DMCs with ≥4 reads within 500 bps to single units requiring ≥eight and ≥4 DMCs for CG and CHN sites, respectively (N = A,T,C,G; H ≠ G). Using these parameters, DMRs were identified across floral bud, early heart, early torpedo, bent cotyledon, mature green and leaf samples, and merged into a single bedFile using the BEDtools *merge* function (*Quinlan and Hall, 2010*). Resulting DMRs were then used to calculate the methylation rate on all analyzed tissues and genotypes. We assigned that a gene and a DMR are associated if the DMR is overlapping within 1.5 kb upstream or downstream of TAIR10 annotated gene bodies using BEDtools *closest* function. For down-regulated genes overlapping with DMRs with above criteria, significance was tested using BEDtools *fisher* function with nuclear genome as a background control.

## mRNA profiling and analysis

Smart-seq2 mRNA libraries were generated from 1 µl of the 7 µl bent cotyledon embryo total RNA as previously described (*Hofmann et al., 2019*; *Picelli et al., 2014*). Both amplified cDNA and final libraries were inspected using Agilent HS NGS Fragment Kit (DNF-474) to control for library quality and proper length distributions. Libraries were sequenced in single-read mode on an Illumina HiSeq 2500 or NextSeq 550 machine. Raw FASTQ files from technical replicates were merged, quality filtered and trimmed for adapter sequences with *Trim Galore* using default parameters. Trimmed reads were aligned using STAR (*Dobin et al., 2013*) against a genome index generated using the TAIR10 genome fasta file and all transcripts in the GTF of Ensembl build TAIR10 annotation set (release version 44). Aligned transcriptome bam files were used to quantify read counts per gene and transcript abundance using RSEM (*Li and Dewey, 2011*). Along with the transcriptomes generated in this study, publicly available embryonic transcriptomes *Hofmann et al., 2019* used for PCA were analyzed in the same fashion as described above (*Supplementary file 6*). Prior to PCA (*Figure 5A*), read counts derived from nuclear protein-coding genes were subjected to variance stabilizing transformations using DESeq2 (*Love et al., 2014*). Differential gene expression analysis was performed using DESeq2 for genes with at least five aligned reads. Genes with ≥2-fold differences and adjusted p-value ≤ 0.01 were classified as differentially expressed genes (DEGs). Nearest-neighbor genes in *Figure 1A,B* were classified based on Euclidean distance. First, the centroid expression of MET1 and VIM1/2/3 was calculated for all tissue types represented in the developmental time series. This centroid value was then used to calculate Euclidean distance of all TAIR10-annotated protein-coding genes and sorted based on their distances.

## ChIP-seq analysis

ChIP-seq data for H2A variants and H3K9me2 were downloaded from GSE50942 (*Yelagandula et al., 2014*) and GSE51304 (*Stroud et al., 2014*), respectively. H3K9 acetylation marks were from GSE98214 (*Wang et al., 2019*). H3K4me3 marks were obtained from GSE152243 (*Maher, 2020*). All FASTQ files were trimmed and quality filtered using *Trim Galore* default parameters. Trimmed reads were aligned against the TAIR10 genome using BWA-MEM (*Li and Durbin, 2009*). Multi-mapping reads and clonal duplicates were removed using *MarkDuplicates* from the Picard Tools suite (*Toolkit, 2019*). The resulting BAM files containing alignments were sorted, indexed and used as input for the *bamCoverage* function of deepTools (*Ramírez et al., 2014*) to obtain genome normalized coverage with parameters −*normalizeUsing* 'RPGC'. Processed bigwig files for H1 Chromatin Affinity purification followed by sequencing (ChAP) and DNase-seq datasets were obtained from GSE122394 (*Choi et al., 2020*). MNase-Seq data was obtained from GSE113556 (*Rutowicz et al., 2019*). ATAC-seq processed bigwig files for WT and *ddm1* mutants were from GSE155503 (*Zhong et al., 2021*).

## Metaplots

ChIP, ATAC, MNase, DNase, and MethylC-seq metaplots were plotted using the R library *Seqplots* (*Stempor and Ahringer, 2016*). Body, upstream, and downstream regions of TEs or genes were split into equal-sized bins, and the average levels for each bin was calculated and plotted.

## CMT3 transgene copy number estimation

CMT3 transgene copy number was estimated using two methods: qPCR and coverage calculation. For the qPCR method, genomic DNA was extracted from leaves of three-week old plants using the CTAB DNA isolation method (*Aboul-Maaty and Oraby, 2019*). Relative transgene copy number was determined by using the qPCR-based method as described (*Shepherd et al., 2009*). *ACTIN2* was used as a control gene while transgene copy number was calculated based on CMT3 levels. For the coverage method, Bismark-aligned and deduplicated BAM files from WT, cCMT3 and rCMT3 lines were processed with DeepTools to obtain normalized genome coverage as bins per million mapped reads (BPM) units with the *bamCoverage* function and following parameters: *–binsize 50 –skipNAs –normalizeUsing* 'BPM' *–ignoreForNormalization mitochondria chloroplast*. The resulting bigwig files were used to calculate genome-wide coverage fold-changes relative to WT using the deepTools function *bigwigCompare –skipNAs –operation 'ratio'.* CMT3 locus was displayed with the Integrative Genomics Viewer (IGV).

## Availability of data and material

All sequencing data generated in this study are available at the National Center for Biotechnology Information Gene Expression Omnibus (NCBI GEO, https://www.ncbi.nlm.nih.gov/geo/) under accession number GSE171198. ChIP-Seq and mRNA-seq bioinformatic analysis pipelines were based on Nextflow (*Di Tommaso et al., 2017*) and the nf-core framework (*Ewels et al., 2020*) are available at https://github.com/Gregor-Mendel-Institute/RKP2021-CMT3, [copy archived at swh:1:rev: 89d7e2ea78af1969bb161640baed09296ed2485f (*Papareddy, 2021*)].

## Acknowledgements

We thank the Vienna Biocenter Core Facilities GmbH (VBCF) Next Generation Sequencing and Plant Sciences Facilities for next-generation sequencing and plant growth chamber access, respectively, and the Institute of Molecular Pathology-Institute of Molecular Biology-Gregor Mendel Institute Molecular Biology Services for instrument access and support. RKP personally thanks Pierre Bourguet and Michael Borg for invaluable discussions. We also thank Zdravko Lorkovic, Bhagyshree Jamge, Robin Burns, Eriko Sasaki, Magnus Nordborg and Frédéric Berger for sharing thoughts and reagents; and members of the Nodine lab for valuable input. This work was supported by the European Research Council under the European Union's Horizon 2020 Research and Innovation Program grant 637888 to MDN.

## Additional information

### Funding

| Funder | Grant reference number | Author |
| --- | --- | --- |
| H2020 European Research Council | 637888 | Michael D Nodine |

The funders had no role in study design, data collection and interpretation, or the decision to submit the work for publication.

### Author contributions

Ranjith K Papareddy, Conceptualization, Software, Formal analysis, Investigation, Methodology, Writing - original draft, Writing - review and editing; Katalin Páldi, Investigation, Methodology; Anna D Smolka, Investigation; Patrick Hüther, Software; Claude Becker, Supervision; Michael D Nodine, Conceptualization, Supervision, Funding acquisition, Writing - original draft, Writing - review and editing

### Author ORCIDs

Claude Becker (iD) http://orcid.org/0000-0003-3406-4670
Michael D Nodine (iD) https://orcid.org/0000-0002-6204-8857

Decision letter and Author response

Decision letter https://doi.org/10.7554/eLife.69396.sa1

Author response https://doi.org/10.7554/eLife.69396.sa2

# Additional files

## Supplementary files

- Supplementary file 1. CG differentially methylated regions during development.
- Supplementary file 2. CHG differentially methylated regions during development.
- Supplementary file 3. Transcript levels in rCMT3 compared to wild-type embryos.
- Supplementary file 4. CHG differentially methylated regions in rCMT3 compared to wild type.
- Supplementary file 5. Oligonucleotides used in this study.
- Supplementary file 6. MethylC-seq and mRNA-seq mapping statistics.
- Transparent reporting form

## Data availability

All sequencing data generated in this study are publicly available at the National Center for Biotechnology Information Gene Expression Omnibus (NCBI GEO, https://www.ncbi.nlm.nih.gov/geo/) under accession number GSE171198.ChIP-Seq and mRNA-seq bioinformatic analysis pipelines were basedon Nextflow and the nf-core framework are available at https://github.com/Gregor-Mendel-Institute/RKP2021-CMT3 (copy archived at https://archive.softwareheritage.org/swh:1:rev: 89d7e2ea78af1969bb161640baed09296ed2485f).

The following dataset was generated:

| Author(s) | Year | Dataset title | Dataset URL | Database and Identifier |
|-----------|------|---------------|-------------|-------------------------|
| Nodine M, Papareddy R | 2021 | Repression of CHROMOMETHYLASE 3 Prevents Epigenetic Collateral Damage in *Arabidopsis* | https://www.ncbi.nlm.nih.gov/geo/query/acc.cgi?acc=GSE171198 | NCBI Gene Expression Omnibus, GSE171198 |

The following previously published datasets were used:

| Author(s) | Year | Dataset title | Dataset URL | Database and Identifier |
|-----------|------|---------------|-------------|-------------------------|
| Nodine M, Hofmann F, Schon M | 2018 | The embryonic transcriptome of *Arabidopsis thaliana* | https://www.ncbi.nlm.nih.gov/geo/query/acc.cgi?acc=GSE121236 | NCBI Gene Expression Omnibus, GSE121236 |
| Nodine M, Papareddy R | 2020 | Chromatin regulates expression of small RNAs to help maintain transposon methylome homeostasis in *Arabidopsis* | https://www.ncbi.nlm.nih.gov/geo/query/acc.cgi?acc=GSE152971 | NCBI Gene Expression Omnibus, GSE152971 |
| Nishimura T, Zilberman D, Nishimura T, Zilberman D | 2012 | Active DNA demethylation in plant companion cells reinforces transposon methylation in gametes | https://www.ncbi.nlm.nih.gov/geo/query/acc.cgi?acc=GSE38935 | NCBI Gene Expression Omnibus, GSE38935 |
| Gehring | 2014 | Natural epigenetic polymorphisms lead to intraspecific variation in *Arabidopsis* gene imprinting | https://www.ncbi.nlm.nih.gov/geo/query/acc.cgi?acc=GSE52814 | NCBI Gene Expression Omnibus, GSE52814 |
| Hsieh T, Ibarra CA, Silva P, Zemach A, Williams LE, Fisher RL, Zilberman D | 2009 | Genome-wide demethylation of *Arabidopsis* endosperm | https://www.ncbi.nlm.nih.gov/geo/query/acc.cgi?acc=GSE15922 | NCBI Gene Expression Omnibus, GSE15922 |
| Bouyer D, Roudier F, Colot V, Bouyer D, Roudier F, Colot V | 2017 | DNA methylation dynamics during early plant life | https://www.ncbi.nlm.nih.gov/geo/query/acc.cgi?acc=GSE85975 | NCBI Gene Expression Omnibus, GSE85975 |
| Stroud H | 2013 | Comprehensive analysis of | https://www.ncbi.nlm. | NCBI Gene |

| | | | | |
|---|---|---|---|---|
| | | silencing mutants reveals complex regulation of the *Arabidopsis* methylome | nih.gov/geo/query/acc. cgi?acc=GSE39901 | Expression Omnibus, GSE39901 |
| Yelagandula R, Stroud H | 2014 | The Histone Variant H2A.W Defines Heterochromatin and Promotes Chromatin Condensation in *Arabidopsis* | https://www.ncbi.nlm. nih.gov/geo/query/acc. cgi?acc=GSE50942 | NCBI Gene Expression Omnibus, GSE50942 |
| Stroud H | 2013 | Non-CG methylation patterns shape the epigenetic landscape in *Arabidopsis* | https://www.ncbi.nlm. nih.gov/geo/query/acc. cgi?acc=GSE51304 | NCBI Gene Expression Omnibus, GSE51304 |
| Wang L, Wang C, Liu X, Cheng J, Li S, Zhu J, Gong Z | 2018 | Peroxisomal *β*-oxidation regulates histone acetylation and DNA methylation in *Arabidopsis* | https://www.ncbi.nlm. nih.gov/geo/query/acc. cgi?acc=GSE98214 | NCBI Gene Expression Omnibus, GSE98214 |
| Maher KA, Wang D, Deal RB, Maher KA, Wang D, Deal RB | 2020 | Differences in directionality of RNA polymerase initiation underlie epigenome differences between plants and animals | https://www.ncbi.nlm. nih.gov/geo/query/acc. cgi?acc=GSE152243 | NCBI Gene Expression Omnibus, GSE152243 |
| Choi J, Lyons DB, Kim MY, Choi J, Lyons DB, Kim MY | 2019 | DNA methylation and histone H1 jointly repress transposable elements and aberrant intragenic transcripts | https://www.ncbi.nlm. nih.gov/geo/query/acc. cgi?acc=GSE122394 | NCBI Gene Expression Omnibus, GSE122394 |
| Lirski M, Kroten MA, Lirski M, Kroten MA | 2019 | MNase analysis of linker histone H1 mutant | https://www.ncbi.nlm. nih.gov/geo/query/acc. cgi?acc=GSE113556 | NCBI Gene Expression Omnibus, GSE113556 |
| Zhenhui Z | 2021 | DNA methylation-linked chromatin accessibility affects genomic architecture in *Arabidopsis* | https://www.ncbi.nlm. nih.gov/geo/query/acc. cgi?acc=GSE155503 | NCBI Gene Expression Omnibus, GSE155503 |

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
