## [Decision Letter]

**Acceptance summary:**

Your findings that the DNA methyltransferase CHROMOMETHYLASE 3 (CMT3) is able to silence transposable elements during plant embryogenesis while not ectopically methylating other genes, and that repression of CMT3 by miR823 during embryogenesis is a key factor in preventing the ectopic accumulation of CHG methylation is quite interesting.

**Decision letter after peer review:**

Thank you for submitting your article "Repression of CHROMOMETHYLASE 3 prevents epigenetic collateral damage in Arabidopsis" for consideration by *eLife*. Your article has been reviewed by 2 peer reviewers, including Richard Amasino as the Reviewing Editor, and the evaluation has been overseen Jürgen Kleine-Vehn as the Senior Editor. Reviewer Hidetoshi Saze has agreed to reveal his identity.

The Reviewing Editor has drafted the following to help you prepare a revised submission.

There are two points that need to be addressed. The first is regarding the data in Figure 4 – providing more detail on how your data relates to that in Wendte et al. 2019 and how you reconcile the ectopic CHGm accumulation patterns in the ddm1 with your model in Figure 6B.

The second is whether or not conclusions like ""CMT3-induced ectopic methylation of genes undergoing transcriptional activation can reduce their corresponding transcript levels" ought to be tempered by a presentation of alternate models and how strong the data actually supports a particular model. We are not asking for more experiments but rather for a more open presentation that will be more informative to the readers of your paper.

*Reviewer #2:*

The manuscript entitled "Repression of CHROMOMETHYLASE3 prevents epigenetic collateral damage in Arabidopsis" demonstrated that transcriptional regulation of CMT3 by miR823 during embryogenesis is essential for preventing the ectopic accumulation of CHG methylation in Arabidopsis. Authors showed that CG and CHG methylation in Arabidopsis embryogenesis is associated with the cell cycle and dynamically regulated. In particular, CHG methylase CMT3 is transcriptionally regulated by miR823. They elegantly showed by genome editing of miR823 and introducing CMT3 transgene resistant to miRNA cleavage that ectopic expression of CMT3 causes accumulation of CHGm in thousands of genes at gene bodies. These genes have "stable" chromatin features. Only a small number of genes associated DMRs showed a reduction in gene expression.

Overall, I found the experiments and data analyses together with genetics and epigenomics were well designed, and the quality of the data is very high. The data about the regulation of CMT3 during the Arabidopsis during embryogenesis is novel and would be of interest to researchers in the relevant fields. Below are specific comments for further improvement of the manuscript.

Figure 4: The results are slightly confusing for interpretation. The ectopic CHGm gain in rCMT3 occurred in genes with pre-existing H3K9me and H2A.W, suggesting that those target genes seem to have heterochromatic features in WT background. A previous study (Wendte et al., 2019) showed that ectopic gain of CHGm occurred in genomic features with pre-existing CHGm, mainly in TE sequences. What kind of genes are enriched, especially in cluster #4? Are they TE-like genes? In addition, ectopic CHGm accumulation in those targets in ddm1 background is inconsistent with your model in Figure 6B. How do you interpret the data?

Figure 5 etc.: The link between ectopic CHGm and gene repression is still weak in the results vis a vis the relatively firm conclusions that "CMT3-induced ectopic methylation of genes undergoing transcriptional activation can reduce their corresponding transcript levels. (Abstract)", and "CMT3-induced hypermethylation can reduce the steady state levels of transcripts from genes.. (L465)". Although thousands of genes accumulate CHGm in rCMT3 lines, a limited number of genes (n=21) associated with DMRs showed reduced expression. The results were consistent with previous studies using ibm1 mutant or CMT3 transgenes, suggesting that CHGm accumulation in gene body does not strongly affect the expression of associated genes. On the other hand, 521 DRG showed reduced expression without CHGm, indicating that the reduction of gene expression is likely due to the secondary effects of rCMT3 expression and other factors. Thus, still, possibilities remain that the 21 genes' repression was also due to side effects of rCMT3 and not to associated CHGm. To prove the direct link between CHGm and gene repression, one would need to perform experiments introducing CHGm to the DMR regions to see whether the induction of CHGm causes repression of the genes. I understand that this is technically difficult even using epigenome editing, and therefore I instead suggest modifying/weaken the statements and conclusions throughout the manuscript.

---

## [Author Response]

There are two points that need to be addressed. The first is regarding the data in Figure 4 – providing more detail on how your data relates to that in Wendte et al. 2019 and how you reconcile the ectopic CHGm accumulation patterns in the ddm1 with your model in Figure 6B.The second is whether or not conclusions like ""CMT3-induced ectopic methylation of genes undergoing transcriptional activation can reduce their corresponding transcript levels" ought to be tempered by a presentation of alternate models and how strong the data actually supports a particular model. We are not asking for more experiments but rather for a more open presentation that will be more informative to the readers of your paper.

We agree with the two major points that were raised. In the revised manuscript, we have included additional text and a supplemental figure to address how our data relates to that in Wendte et al. 2019, as well as how we interpret the ectopic CHG methylation observed in *ddm1* according to our model in Figure 6B. Please find more details below in response to Reviewer #2’s comment.

Regarding the conclusion that "CMT3-induced ectopic methylation of genes undergoing transcriptional activation can reduce their corresponding transcript levels", we agree that alternate models also exist and additional experiments are required to directly test the relationship between CHG methylation of genes and their expression. We have attempted to make this more clear and weakened the strength of this conclusion in the revised manuscript. We have documented the specific changes below in response to Reviewer #2’s comment.

Reviewer #2:The manuscript entitled "Repression of CHROMOMETHYLASE3 prevents epigenetic collateral damage in Arabidopsis" demonstrated that transcriptional regulation of CMT3 by miR823 during embryogenesis is essential for preventing the ectopic accumulation of CHG methylation in Arabidopsis. Authors showed that CG and CHG methylation in Arabidopsis embryogenesis is associated with the cell cycle and dynamically regulated. In particular, CHG methylase CMT3 is transcriptionally regulated by miR823. They elegantly showed by genome editing of miR823 and introducing CMT3 transgene resistant to miRNA cleavage that ectopic expression of CMT3 causes accumulation of CHGm in thousands of genes at gene bodies. These genes have "stable" chromatin features. Only a small number of genes associated DMRs showed a reduction in gene expression.Overall, I found the experiments and data analyses together with genetics and epigenomics were well designed, and the quality of the data is very high. The data about the regulation of CMT3 during the Arabidopsis during embryogenesis is novel and would be of interest to researchers in the relevant fields. Below are specific comments for further improvement of the manuscript.Figure 4: The results are slightly confusing for interpretation. The ectopic CHGm gain in rCMT3 occurred in genes with pre-existing H3K9me and H2A.W, suggesting that those target genes seem to have heterochromatic features in WT background. A previous study (Wendte et al., 2019) showed that ectopic gain of CHGm occurred in genomic features with pre-existing CHGm, mainly in TE sequences. What kind of genes are enriched, especially in cluster #4? Are they TE-like genes? In addition, ectopic CHGm accumulation in those targets in ddm1 background is inconsistent with your model in Figure 6B. How do you interpret the data?

Because misannotation of TEs as genes can confound our analysis we filtered for genes with expression at least >= 1 TPM in at least of the developmental stages (n = 22,637 based on Hofmann et al. 2019). TE-like methylation patterns (teMs) are characterized by enrichment of non-CG methylation, particularly mCHG on their gene bodies, without strong 3’ bias (Kawakatsu et al. 2016). To further test whether mis-annotated TEs complicates our analysis, we have also overlapped our gene clusters with all genes that are classified as teMs in *Arabidopsis thaliana* by Bewick et al. 2016. We found that only 0.01% of teMs overlapped the 22,637 genes that we used for clustering analyses (Figure 4—figure supplement 1D) suggesting that our analysis was not confounded by teMs. However, the number of teMs increased between clusters 1 and 4, as expected because cluster #4 is closer to centromeric regions (Figure 4—figure supplement 1E,N). However, even after excluding the 272 genes that are classified as teMs, we still observed increased CHG hypermethylation in higher numbered clusters (i.e. 4 >> 3 >> 2 > 1) in rCMT3 embryos whereas WT embryos are unmethylated in all clusters (Figure 4—figure supplement 1F,G). Altogether, this indicates that ectopic mCHG in genes is not confounded by teMs and we have included this new analysis in the revised manuscript.

We think that the ectopic accumulation of CHG methylation in *ddm1* mutants (Figure 4J) is consistent with our model (Figure 6B). Upon heterochromatin destabilization in *ddm1* mutants (Figure 4K), CMT3 and other factors are not as strongly targeted to heterochromatic regions and thus can ectopically methylate less heterochromatic genic regions such as those in cluster #4 that more readily attract CMT3 due to features associated with stable nucleosomes. We hope the text that we included on lines 585-597 and the legend of Figure 6B makes it clear that the ectopic gain of CHG methylation in *ddm1* mutants is consistent with our model.

Figure 5 etc.: The link between ectopic CHGm and gene repression is still weak in the results vis a vis the relatively firm conclusions that "CMT3-induced ectopic methylation of genes undergoing transcriptional activation can reduce their corresponding transcript levels. (Abstract)", and "CMT3-induced hypermethylation can reduce the steady state levels of transcripts from genes.. (L465)". Although thousands of genes accumulate CHGm in rCMT3 lines, a limited number of genes (n=21) associated with DMRs showed reduced expression. The results were consistent with previous studies using ibm1 mutant or CMT3 transgenes, suggesting that CHGm accumulation in gene body does not strongly affect the expression of associated genes. On the other hand, 521 DRG showed reduced expression without CHGm, indicating that the reduction of gene expression is likely due to the secondary effects of rCMT3 expression and other factors. Thus, still, possibilities remain that the 21 genes' repression was also due to side effects of rCMT3 and not to associated CHGm. To prove the direct link between CHGm and gene repression, one would need to perform experiments introducing CHGm to the DMR regions to see whether the induction of CHGm causes repression of the genes. I understand that this is technically difficult even using epigenome editing, and therefore I instead suggest modifying/weaken the statements and conclusions throughout the manuscript.

Thank you for this comment and we agree that it is difficult to directly link ectopic CHG methylation based on our current results. More experiments such as introducing CHG methylation to DMRs in a wild-type background would be required to directly test the model we proposed. However, we agree that these experiments are currently technically difficult and beyond the scope of the current study. Therefore, we have modified and weakened the statements regarding the link between CHG methylation and gene repression as described below.

We agree that we cannot conclude that the down-regulation of hypermethylated genes is strictly due to gene bodies or promoter regions. We have attempted to further clarify this in the revised abstract.

Also, it is true that 521 DRGs are not DMRs and agree that we cannot exclude that DEGs are due to secondary effects as we stated on lines 469-472:

“In both rCMT3 lines, DEGs were less hypermethylated compared to all expressed genes, which indicates that the vast majority of changes in gene expression observed upon up-regulation of CMT3 was not directly due to their hypermethylation (Figure 5—figure supplement 1D).”

However, the 21 down-regulated genes that did overlap DMRs were significantly hypermethylated in embryos expressing the miR823-resistant CMT3 transgenes. The amount of hypermethylation of these 21 genes was also significantly higher in the more strongly expressing CMT3 transgenic line #3 compared to line #1. We think this is consistent with CMT3-induced hypermethylation repressing gene expression. We agree that additional experiments would be required to test whether gene-body CHG methylation can repress gene expression and have added text to the sentences on lines 499-500 of the revised manuscript in an attempt to make this more clear.

It is true that CHG methylation generally does not cause a reduction of steady-state transcript levels. Perhaps we were fortunate to be measuring transcript levels at a developmental stage in which a large set of genes are rapidly activated during the onset of embryo maturation. We have added this point to the Discussion (lines 606-609), as well as stated that additional experiments are required “to directly test the relationship between CMT3-induced hypermethylation and gene expression” on lines 609-611 of the revised text. We also weakened a similar concluding statement on lines 545 and 615-616 of the revised text.